



# A globally-applicable framework for compound flood hazard modeling

Dirk Eilander[1,2], Anaïs Couasnon[1], Tim Leijnse[2], Hiroaki Ikeuchi[3], Dai Yamazaki[4], Sanne Muis[1,2], Job Dullaart[1], Hessel C. Winsemius[2], Philip J. Ward[1]

[1]Institute for Environmental Studies (IVM), Vrije Universiteit Amsterdam, Amsterdam, The Netherlands
[2]Deltares, Delft, The Netherlands
[3]Ministry of Land, Infrastructure, Transport and Tourism, Tokyo, Japan
[4]Institute of Industrial Sciences, the University of Tokyo, Tokyo, Japan

*Correspondence to*: Dirk Eilander (dirk.eilander@deltares.nl)

**Abstract.** Coastal river deltas are susceptible to flooding from pluvial, fluvial, and coastal flood drivers. Compound floods, which result from the co-occurrence of two or more of these drivers, typically exacerbate impacts compared to floods from a single driver. While several global flood models have been developed, these do not account for compound flooding. Local scale compound flood models provide state-of-the-art analyses but are hard to scale up as these typically are based on local

datasets. Hence, there is a need for globally-applicable compound flood hazard modeling. We develop, validate and apply a framework for compound flood hazard modeling, which consists of the local high-resolution 2D hydrodynamic flood model SFINCS, which is automatically set up from global datasets and loosely coupled with a global hydrodynamic river routing and flood model, as well as a global surge and tide model to account for interactions between all drivers. To test the framework, we simulate two historical compound flood events, cyclones Idai and Eloise, in the Sofala province of

Mozambique, and compare the flood extent to observations from remote sensing and to the global quasi 2D CaMa-Flood model. The results show that while the global and local model have similar skill in terms of the critical success index, they result in rather different flood maps. On the one hand, the local model has a higher hit ratio due to the representation of direct coastal and pluvial flooding (rain on grid) and a higher floodplain connectivity. It also shows a faster response to coastal drivers within the estuaries and more realistic flood depth maps. On the other hand, the local model has a higher false

alarm ratio, which is partly explained by the inclusion of direct pluvial flooding without sufficient representation of small scale (subgrid) drainage capacity. To showcase a possible application of the framework, we also determine the dominant flood drivers and transition zones between flood drivers for both events. These vary significantly between both events because of differences in the magnitude of and time lag between the flood drivers. We argue that a wide range of plausible events should be investigated to get a robust understanding of compound flood interactions, which is important to understand

for flood adaptation, preparedness, and response. As the model setup and coupling is automated, reproducible, and globally applicable, the presented framework is a promising step forward towards large scale compound flood hazard modeling.



## 1. Introduction

Coastal river deltas are susceptible to flooding due to their physical setting in low elevation regions and the presence of many densely populated cities. A recent study showed that deltas contain 4.5% of the global population in 2017, while only covering 0.57% of the earth's land surface area (Edmonds et al., 2020). Floods in these delta regions can occur as the result of different physical drivers, including extreme rainfall, river discharge, or extreme coastal water levels. Floods can also occur (or be exacerbated) by the co-occurrence of combinations of these drivers, so-called compound flood events, which may amplify the total flood hazard (Leonard et al., 2014; Zscheischler et al., 2018). Tropical cyclone Idai, which made landfall near Beira, Mozambique in March 2019, caused more than 600 casualties and affected an estimated 1.85 million people (UN OCHA, 2019). This is an example of the devastating impacts that compound floods can cause (Emerton et al., 2020). A comprehensive understanding of flood risk in deltas is therefore crucial for effective risk reduction.

There is a wide recognition that it is important to take interactions between flood drivers into account for flood risk assessment and management in both the scientific (Moftakhari et al., 2017; Wahl et al., 2015; Ward et al., 2018) and decision-making (Browder et al., 2021; UNDRR, 2019) communities. Several studies have used statistical models to assess the dependence between flood drivers in order to understand the likelihood of extreme drivers occurring together (Camus et al., 2021; Couasnon et al., 2020; Bevacqua et al., 2019; Hendry et al., 2019; Ward et al., 2018). Furthermore, hydrodynamic model simulations have been used to understand the complex physical interactions between drivers and their relative importance for the total flood hazard (Bakhtyar et al., 2020; Eilander et al., 2020; Gori et al., 2020a; Harrison et al., 2021; Kumbier et al., 2018; Muñoz et al., 2021; Olbert et al., 2017; Santiago-Collazo et al., 2019; Torres et al., 2015)

However, to date most global flood risk models still analyze each flood driver in isolation (Alfieri et al., 2017; Hirabayashi et al., 2021; Tiggeloven et al., 2020; Vousdoukas et al., 2018; Ward et al., 2020). Recently, the effect of storm surge on fluvial flooding was analyzed at the global scale, showing that 1-in-10 years fluvial flood levels are exacerbated by surge for 64% of the locations analyzed, causing increased flood risk for 9.3% of the population exposed to riverine flooding (Eilander et al., 2020; Ikeuchi et al., 2017). Bates et al. (2021) were the first to make a combined risk assessment of fluvial, pluvial and coastal flood hazard for the continental US, but did not account for physical interactions of pluvial with other flood drivers.

While the performance and resolution of large-scale flood models is approaching that of local-scale flood models in data-rich areas (Wing et al., 2019), there are still large differences between global flood models in many areas globally (Aerts et al., 2020; Bernhofen et al., 2018; Trigg et al., 2016). The setup of these models remains a challenging task, due to the lack of open and accurate high-resolution global topography data (Hawker et al., 2018b) as well as missing data on river and estuarine bathymetry (Neal et al., 2021) and flood defenses (Ward et al., 2015; Wing et al., 2019). Therefore, building hydrodynamic flood models from global datasets requires several data preprocessing steps that may have a large effect on the model skill (Sampson et al., 2015). Furthermore, the code for setting up most global flood models is closed source, while

 

an open source framework would increase the comparability and reproducibility by providing a transparent workflow (Hall et al., 2021; Hoch and Trigg, 2019). Sosa et al. (2020) presented an automatic model builder for LISFLOOD-FP models, and

Uhe et al. (2021) extended this framework to a model cascade to compute fluvial flood hazard from meteorological drivers. Van Ormondt et al. (2020) developed Delft Dashboard, which is a graphical user interface with various modular toolboxes to semi-automatically setup hydrodynamic models schematizations in the ocean and coastal domains, but lacks tools to couple riverine models. This leaves a gap for a fully automated model builder that can be applied to the complex coastal delta environment to simulate compound flood events.

In this study we present an automated framework to model compound flooding anywhere on the globe in a reproducible and transparent manner. The framework consists of a local hydrodynamic model, which is automatically built from global datasets and loosely coupled with a global hydrological and river routing model for upstream boundary conditions and a global surge and tide model for downstream boundary conditions. The goal of this study is to present the framework and to test its ability to simulate compound floods in data-sparse coastal deltas. In particular, we compare flood hazard maps from

the local hydrodynamic model against satellite observations of flood extents for two historical events. To evaluate the added value of using the local model, we also compare against a global model. Furthermore, to demonstrate a potential application of the framework, we identify main flood drivers and transition zones between drivers following Bilskie and Hagen (2018), which is important information for flood adaptation, preparedness and response.

## 2. Case study

To evaluate the flood hazard framework, we apply it to two historical events in the Sofala province of Mozambique, namely tropical cyclone Idai in March 2019 and tropical cyclone Eloise in January 2021. Both events are examples of compound flood events in a coastal delta. Due to the relative data scarcity, The Global Flood Awareness System (GloFAS) has been shown to be useful in supporting decision making in this area (Emerton et al., 2020). The largest city in the Sofala province is Beira, with more than 500,000 inhabitants and a large port connecting the hinterland with the Indian Ocean. While the city

itself is mainly threatened by coastal and pluvial flooding, the deltas of the Pungwe and Buzi rivers are also susceptible to fluvial flooding (Emerton et al., 2020; van Berchum et al., 2020).

Tropical cyclone Idai originated in the Mozambique Channel as a tropical depression, which already caused extensive flooding after its first landfall in early March. After it moved back over the Mozambique Channel it gained intensity and became a tropical cyclone with 10-min sustained wind speeds of 165 km/h, a maximum calculated surge of ~4.4 m, and

torrential rainfall during the second landfall near Beira on 15th March (ERCC, 2019). After the second landfall, large areas flooded, first around the coast, followed a few days later by the Buzi and Pungwe floodplains. The tropical cyclone destroyed more than 60,000 houses and an estimated 286,000 people received shelter (UN OCHA, 2019).



Tropical cyclone Eloise made landfall on 23 January 2021 around 20 km south of Beira, with winds of 140 km/h and widespread and extreme rainfall. The region experienced widespread post-cyclone flooding while it was already hit by heavy rainfall on January 15 and subsequent high river water levels and was still recovering from the 2019 flood after tropical cyclone Idai. The Sofala province was the most affected, especially communities along the Pungwe and Buzi rivers. In total, more than 8,800 houses were damaged and 176,000 people were affected (UN OCHA, 2021).

## 3. Methods

The globally-applicable compound flood hazard framework is shown in Figure 1. In Section 3.1 we describe the **global models** used to set the boundary conditions of the local model. In Section 3.2 we discuss the **local hydrodynamic model** SFINCS as well as its automated setup. In Section 3.3 we discuss the **analysis** of the model results and the compound flood drivers. Both the model setup and analysis (post-processing) are facilitated through HydroMT v0.4.5 (Eilander and Boisgontier, 2022), an open-source Python package to automate the building and analysis of geoscientific models, and the model specific SFINCS plugin HydroMT-SFINCS v0.2.1 (Eilander et al., 2022). All required model pre- and postprocessing steps have been automated and can thus easily be repeated for different locations. The approach is modular as datasets can easily be interchanged, also for higher resolution local datasets if available, and many workflows to process raw data into model input data can be reused for different models.

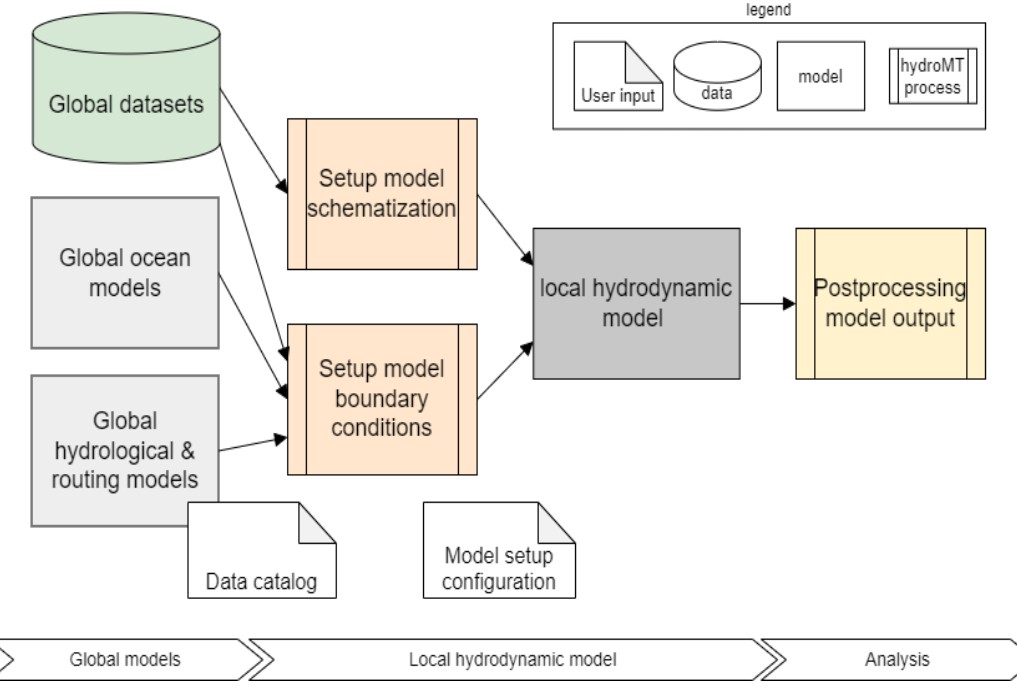

**Figure 1: Framework for globally applicable compound flood hazard modeling**



## 3.1 Global models

To make the framework globally applicable, we make use of global models to force the local flood model. The following sections describe the global ocean models used for the coastal boundary conditions and global hydrological and routing models used for the fluvial boundary conditions. To ensure coherence between the flood drivers, the atmospheric forcing of all models is based on the ERA5 reanalysis dataset, which has a 15 arcmin spatial resolution (~31 km at the equator) and a 1 hour temporal resolution (Hersbach et al., 2020).

### 3.1.1 Global ocean models

Total nearshore water levels consist of several components, namely astronomical tide, storm surge and wave setup. The latter two are episodic fluctuations due to atmospheric drivers. Storm surge is generated by a storm's winds pushing water onshore and the inverted barometer effects of the pressure (Resio and Westerink, 2008). Wave setup is an episodic wave-driven increase of nearshore water levels resulting from wave shoaling and breaking processes (Bowen et al., 1968).

The tide and surge components are simulated with the Global Tide and Surge Model (GTSM) version 3.0 (Muis et al., 2020) which is based on the Delft3D Flexible Mesh hydrodynamic model software (Kernkamp et al., 2011). The model resolution varies from 25 km in the deep ocean to 2.5 km (1.25 km in Europe) near the coast and results are stored at a 10 min temporal resolution. Details about the GTSM model schematization and parameterization are discussed in Muis et al. (2020) and Wang et al. (2021). In this study GTSM is forced with mean sea level pressure and 10m meridional (v; northward) and zonal (u; eastward) wind components from ERA5 merged with wind and pressure fields from the Holland parametric wind model (Holland, 1980) based on the IBTrACS (Knapp et al., 2010). The data from the Holland model are described with a polar grid with 36 radial and a radius of 750 km following Dullaart et al. (2021), where the data in the outermost 33% are linearly interpolated with the background ERA5 data to avoid a wind speed and pressure drop towards the outer rim (Deltares, 2022, April 2). The simulated tides are based on tide-generating forces at 60 frequencies without assimilation of satellite altimetry (Irazoqui Apecechea et al., 2017). The GTSM model has been validated for various historical hurricane events (Dullaart et al., 2020) and for derived return levels (Muis et al., 2020), showing good agreement with observations.

Hourly time series of significant height of wind waves ($Hs$) are extracted at GTSM output locations from the 30 arcmin ERA5 dataset (Bidlot, 2012; Hersbach et al., 2020). We estimate the wave setup component based on $0.2Hs$, which is an often used approximation for (large-scale) studies (Camus et al., 2021; Vousdoukas et al., 2016; US Army Corps of Engineers, 2002). Time series of total water level ($Htwl$) are derived by combining the GTSM tide and storm surge components ($Hst$) with the wave setup component: $Htwl = Hst + 0.2Hs$, where $Hs$ is linearly interpolated to 10 min intervals to match the GTSM temporal resolution.



Due to a lack of observations from coastal water level gauges, a quantitative validation of the simulated water levels is not possible, but some general observations about the simulated data can be made. The maximum water levels during both events (5.0 m+MSL during Idai; 3.8 m+MSL during Eloise) occurred close to neap tide and are caused by surge (3.2 m during Idai; 2.6 m during Eloise) and wind setup (1.2 m during Idai; 0.7 m during Eloise). The maximum surge and its timing during Idai (4.0 m) is in line with the operational forecast of 4.4 m (ERCC, 2019). In comparison with IHO tidal constituents as retrieved using the Delft Dashboard (van Ormondt et al., 2020), the highest astronomical tide near Beira is expected to be around 3.8 m while our simulations result in 4.5 m, indicating an overestimation. This has however little effect on the maximum water levels which occurred close to neap tide.

### 3.1.2 Global hydrological & routing models

Riverine discharge is simulated with the global river routing modeling CaMa-Flood version 4.0.1 (Yamazaki et al., 2013; Hirabayashi et al., 2021). CaMa-Flood is selected as to our knowledge it is the only global river routing model with an explicit representation of floodplains (Zhao et al., 2017) that also accounts for downstream sea level boundary conditions (Ikeuchi et al., 2017). CaMa-Flood uses the local inertial approximation (Bates et al., 2010) to solve the mass and momentum equations for river and floodplain flows in one dimension (Yamazaki et al., 2013). A model grid cell represents a unit catchment containing a river segment with a rectangular cross section and a floodplain profile based on subgrid topography. In CaMa-Flood version v4.0 and later the subgrid parameters are based on the global high-resolution topography data MERIT DEM (Yamazaki et al., 2017) and hydrography data MERIT Hydro (Yamazaki et al., 2019). Each river segment is connected to one downstream neighbor, but floodplains of neighboring unit catchments can exchange flows through so-called bifurcation channels, making it a quasi 2D model. The bifurcation channels are based on a set number of elevation thresholds for which a representative stream width at the interface between the floodplains of two neighboring unit catchments is derived based on the subgrid topography. Bifurcation channels are activated if the surface water elevation exceeds an elevation threshold. These bifurcation channels are shown to be important in flat coastal areas to correctly simulate floodplain connectivity (Ikeuchi et al., 2015; Mateo et al., 2017; Yamazaki et al., 2014). The unit-catchment areas are used to interpolate the input runoff to the model grid, where the runoff within the unit-catchment directly enters the river segment at its upstream end.

We use a regional cutout between 32 °W, -21.5 °S, 35.5 °E and -17 °N of the 3 arcmin spatial resolution global CaMa-Flood schematization, see Figure 2A. Default model settings are used except for the bifurcation scheme, which is defined at 10 instead of 5 elevation thresholds to maximize floodplain connectivity. Furthermore, to make the model comparable with the local model river width and depth maps are created using the same procedure as explained in Section 3.2.1, but with the CaMa-Flood river segments. CaMa-Flood is forced with ERA5 runoff, which is simulated with the Hydrological Tiled ECMWF Scheme for Surface Exchanges over Land (HTESSEL) (Balsamo et al., 2009), and total sea water levels from the nearest GTSM output location at all river outlet locations, see Section 3.1.1. Grids of instantaneous discharge and flood



depth with a daily temporal resolution are saved to be used as input for the local flood model. The flood depth maps at the

model resolution are downscaled to a 3 arsec (~100 m at the equator) resolution based on high resolution topography.

During tropical cyclone Idai, national hydrological bulletins reported water levels for the Pungwe river at Mefambisse and

for the Buzi river at Goonda (approximate locations are shown in Figure 2A). The bulletins report water levels during the

onset and recession of the flood peak but missed the peak itself. Furthermore, neither exact locations nor the used vertical

reference level could be retrieved, making a quantitative comparison impossible. We therefore only make a qualitative

comparison between the observed and with CaMa-Flood simulated water levels. Compared to the observations, the

simulated flood peak at the Pungwe river is slightly delayed but seems to correctly capture the recession, while the flood

peak at the Buzi river seems to be overestimated and the recession too fast (Figure A1). The overestimation could be the

result of missing schematization of reservoirs in the model, such as the Chicamba reservoir in the Revue river, a tributary to

the Buzi river.

## 3.2 Local hydrodynamic model

The Super-Fast INundation of CoastS (SFINCS) model (Leijnse et al., 2021) is used to simulate water levels and overland

flood depths within coastal deltas. SFINCS is selected as it is designed to efficiently simulate overland flow from compound

flooding at limited computation costs and with good accuracy (Leijnse et al., 2021; Sebastian et al., 2021). The governing

equations of the SFINCS model are based on the local inertia equations in two dimensions (Bates et al., 2010). First, the flow

rate is solved based on two 1D momentum equations in x and y directions with spatially varying roughness. Then, the water

levels are computed based on the mass balance. On-grid precipitation and discharge boundary conditions are added as a local

source term in the mass equation. At open boundaries, the model is forced with dynamic water levels, which are interpolated

from the nearest user defined point location with water levels. For a full description of the model we refer the reader to

Leijnse et al. (2021). Here we use the SFINCS code revision 295.

In the remainder of this section we describe the steps taken to automatically setup the SFINCS model schematization and

forcing from global datasets using HydroMT-SFINCS v0.2.1 (Eilander et al., 2022). The complete model setup process is

described in a single configuration *ini* file, and all datasets (see Table 1) in a single data catalog *yaml* file, see appendix B.

This improves the transparency and reproducibility of the model setup.

**Table 1: Overview of global datasets used to setup the local flood model**

| Dataset | Variable [units] |
|---|---|
| ERA5 (Hersbach et al., 2020) | Total Runoff (ro) [m/hr] |





| MERIT Hydro (Yamazaki et al., 2019) | Elevation [m+EGM96] |
|---|---|
| | Upstream area [m$^2$] |
| | D8 flow directions [-] |
| GRWL (Allen and Pavelsky, 2018) | Permanent water mask [-] |
| River width datasets (Lin et al., 2020) | River width [m] |
| | Bankfull discharge [m$^3$/s] |
| CNES-CLS18  (Mulet et al., 2021) | Mean dynamic topography [m] |
| OSM ocean shapefile (Coastline data sets, 2020) | Ocean shapefile [-] |

### 3.2.1 Setup model schematization

*Step 1: Model grid definition*

The SFINCS model grid is set up based on a bounding box of the area of interest, a resolution and a projected coordinate reference system, here between 34.33 °W, -20.12 °S, 34.95 °E and -19.30 °N (WGS84) at 100 m resolution in UTM zone 36S projection.

*Step 2: Topography and hydrography data*

Topography data is reprojected to the model grid using bilinear interpolation. As hydrography data (D8 flow directions and upstream area) cannot be reprojected directly, we instead reproject a pseudo-topography grid based on upstream area and subsequently derive flow directions. The upstream area is then recalculated based on the new flow directions taking into account the upstream area of inflowing rivers and streams at the model domain boundary. The hydrography maps are not

used by SFINCS but used at later stages of the automatic model setup to define river bathymetry and river in- and outflow locations. Here we used topography and hydrography data from MERIT Hydro v1.0 (Yamazaki et al., 2019).

*Step 3: River and estuarine bathymetry*

As global digital elevation models (DEMs) do not represent the bed level of river channels, the river bathymetry is burned

into the data using a similar procedure as in Sampson et al. (2015). Rivers are defined based on an upstream area threshold and discretized into river segments. For each segment, we first determine the river width from a binary river mask, then the river bankfull elevation from the cells adjacent to the river mask and finally the river depth relative to the bankfull elevation. The detailed procedure is explained here.

- Rivers are based on D8 flow directions and a minimal upstream area threshold. River segments are defined between
225       river confluences or a river headwater cell or outlet cell and a confluence. Long segments are split into equal parts





to approximate a user defined length. Here, we used a minimal upstream area threshold of 100 km² and an approximate segment length of 5 km.

- The river width is calculated as the segment average width derived from a binary river mask, by dividing the surface area of each segment by its length, where the areas across multiple parallel estuarine channels are summed. The mask is primarily based on the Global River Widths from Landsat (GRWL) Database (Allen and Pavelsky, 2018), but extended by rasterizing the river width from the Lin et al. (2020) dataset. The Lin dataset contains river width estimates for ~1.6 km river segments based on a machine learning approach that uses 16 covariates and was trained based on an average width from GRWL and MERIT Hydro. Compared to MERIT Hydro or GRWL it has a higher spatial coverage and extends to smaller rivers with a minimum width of 30 m.

- The river bankfull elevation, relative to the segment elevation, is estimated from a low percentile of height above the nearest river values of cells neighboring the river mask. These values are then corrected such that the absolute bankfull elevation levels are monotonically increasing in upstream direction using the algorithm developed by Yamazaki et al. (2012). Here we use the 25th percentile, which was found to give good results for this region but might need to be refined for other regions.

- We distinguish between a fluvial and estuarine part of the river to determine the river depth. The riverine depth $h$ [m] is estimated from the bankfull discharge Q [m³s⁻¹] using a power-law relationship: $h = aQ^b$, where the default values for $a$ (0.27) and $b$ (0.30) are based on Andreadis et al. (2013) The bankfull discharge is based on the 1-in-2 year return values of the discharge as simulated by Lin et al. (2019), and derived from the nearest river segment from the Lin et al. (2020) dataset. Gaps in bankfull discharge data are filled based on the nearest valid upstream value. The estuarine depth is kept constant based on the depth of the most upstream estuarine segment, which provides a first-order approximation of the depth in ungauged estuaries and is in accordance with observed depths in ideal alluvial estuaries in low-gradient regions (Gisen and Savenije, 2015). Estuarine segments are classified based on a width convergence rate. Natural alluvial estuaries have a funnel planform shape that is wide at the ocean and narrows inland (Savenije, 2005). Here we use a convergence rate threshold of 0.01 m/m applied to a smoothed segment average width. This value was found based on trial and error for the estuaries under consideration and might need to be refined for other locations. A global minimum river depth of 0.5 m is used.

- The river bed elevation $zb$ [m+EGM96] is calculated for each model cell of a river segment from the cell elevation $z0$ [m+EGM96], relative bankfull elevation difference $dz$ [m] and the bankfull depth $h$ [m]: $zb = z0 + max(0, dz - h)$. This bed level is burned into the river center cells and spread to neighboring cells within the river mask to burn a rectangular river profile in the DEM. Finally, we ensure that each river cell has at least one horizontally or vertically neighboring cell with the same or lower elevation to ensure the river has D4 connectivity in the model.





*Step 4: Manning roughness*

A spatially varying manning roughness grid is set up that differentiates between land and river cells, based on the river mask as defined in the previous step. Here we used a constant of 0.03 $sm^{-1/3}$ for river cells and 0.1 $sm^{-1/3}$ for land cells, which is in line with other studies (e.g. Di Baldassarre et al., 2009) and consistent with the global CaMa-Flood model (Yamazaki et al., 2011). HydroMT-SFINCS also contains a routine to set up a spatially varying roughness grid based on land-cover data which is not used here to keep the model consistent with CaMa-Flood.

*Step 5: Boundary cells*

By default, the cells at the edge of the model domain have closed boundaries, but these can be changed to Riemann-type open water level boundaries. Here, an open water level boundary is set for all cells at the interface with the ocean by intersecting the model domain edge cells with the OSM ocean shapefile (Coastline data sets, 2020). In the absence of water level forcing of rivers leaving the model domain at the south and east model boundaries, and to avoid water building up within the model domain, open boundary cells with a zero water depth are set at these locations. These open boundary cells are derived from the previously set hydrography data based on a user-defined upstream area threshold and a river width, here 10 $km^2$ and 1 km respectively.

*Step 6: River inflow points*

Discharge boundary conditions are set at source point locations within the model domain. These points are based on cells where a river flows into the model domain. Rivers are based on a user-defined upstream area and river length thresholds and derived from the hydrography data as derived previously. The minimum length threshold is used to filter short river segments that flow in and out of the model domain. Here we use an upstream area threshold of 500 $km^2$ and minimum length of 10 km to force the model with discharge from the four largest rivers flowing into the model domain, see Figure 2B.



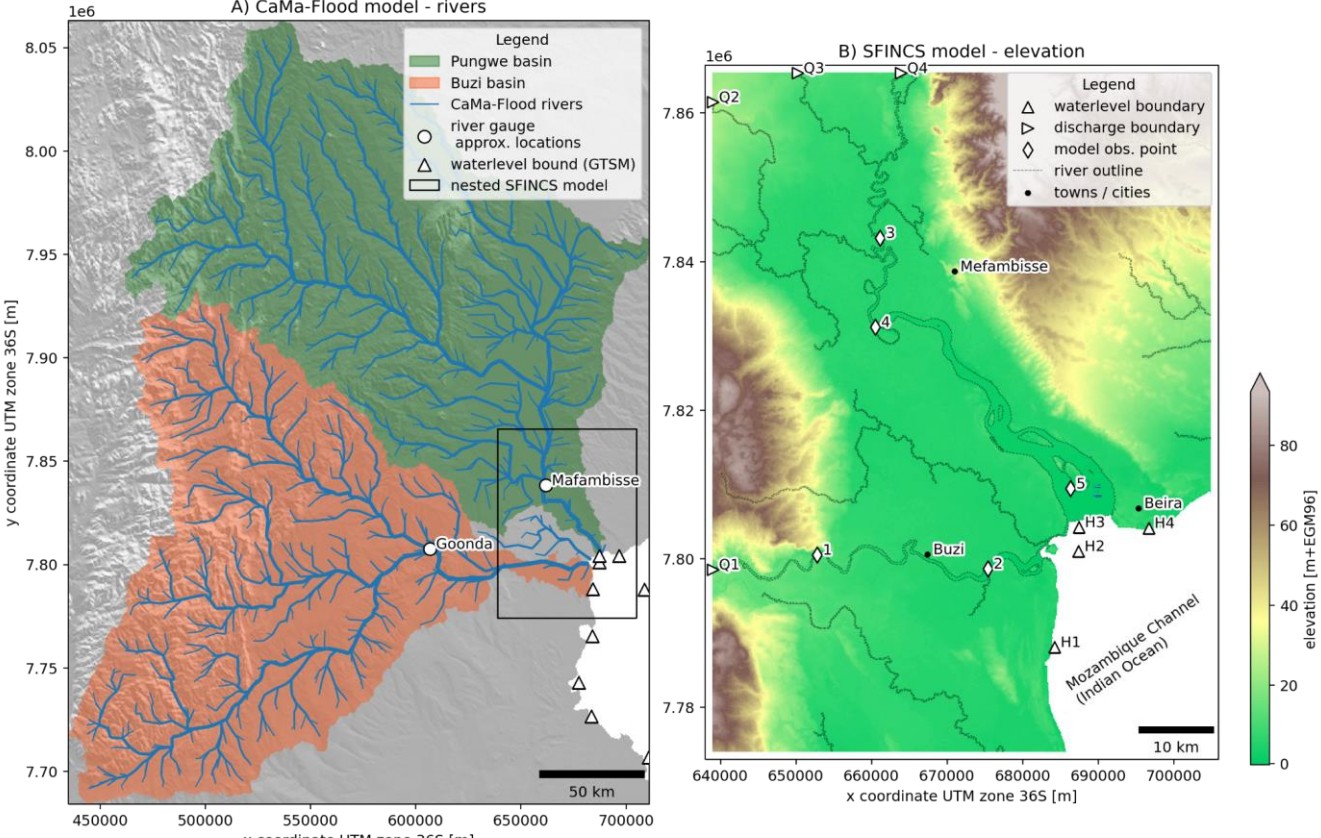

**Figure 2: Maps of a regional cutout of the global CaMa-Flood model (left panel); and the local SFINCS model with boundary condition and model observation locations for the case study in Sofala province, Mozambique (right panel). Note that both maps are in different projections based on the projection used for the model schematization.**

### 3.2.2 Setup model boundary conditions

The following steps, dealing with dynamic boundary conditions, are repeated for each event and/or sensitivity scenario (see Section 3.3.3). The model boundary conditions for both historical events are shown in Figure 3.

*Step 7: Coastal boundary*

Water level boundary conditions are defined at point locations and interpolated by SFINCS to the nearest water level boundary cell. Water level data for the model simulation time period are selected from (global) water level point time series datasets based on a maximum distance from the water level boundary cells (step 5 in Section 3.2.1). The water level data can optionally be corrected for the offset between the vertical datum of the water level and topography data. Here, we use a maximum distance of 5 km to select GTSM output locations and correct these for the difference between MSL and the EGM96 vertical datum based on the CNES-CLS18 mean dynamic topography (Mulet et al., 2021). Note that this offset





amounts to ~0.8 m on average for the selected output locations. The total water level time series at a representative location for both events are shown in the top panels of Figure 3 (full line).

300    *Step 8: Fluvial boundary*

Discharge boundary conditions are defined at source point locations (step 6 in Section 3.2.1) within the model domain. Discharge data for the simulation time period are selected from a gridded discharge dataset. As the (global) discharge dataset is typically based on another (coarser resolution) river network, the source point locations must be matched with a corresponding river cell, which is not necessarily at the exact same location. A matching river cell is defined as the cell

305    within a user-defined maximum search radius that has the smallest difference in upstream area with the inflow point location, that is at least smaller than a user-defined threshold for the absolute or relative difference. Here, we select discharge from the gridded CaMa-Flood model output within a 1 cell search window around the source point location based on a maximum relative error of 5% or absolute error of 100 km$^2$. The discharge time series at the two main rivers for both events are shown in the center panels of Figure 3 (full line).

310

*Step 9: Pluvial boundary*

We use spatially varying precipitation fields for direct rainfall-on-grid forcing. The data are derived from (global) gridded precipitation datasets for the model domain and simulation time period and reprojected to the model projected coordinate system in a resolution similar to the source resolution. Here we use ERA5 runoff rather than precipitation to account for

315    hydrological processes such as infiltration and evaporation and to ensure comparability with the global CaMa-Flood model. Note that infiltration can also be simulated within SFINCS but was turned off for this experiment as this process is accounted for by using runoff instead of precipitation data. The spatially average runoff time series for both events are shown in the bottom panels of Figure 3.





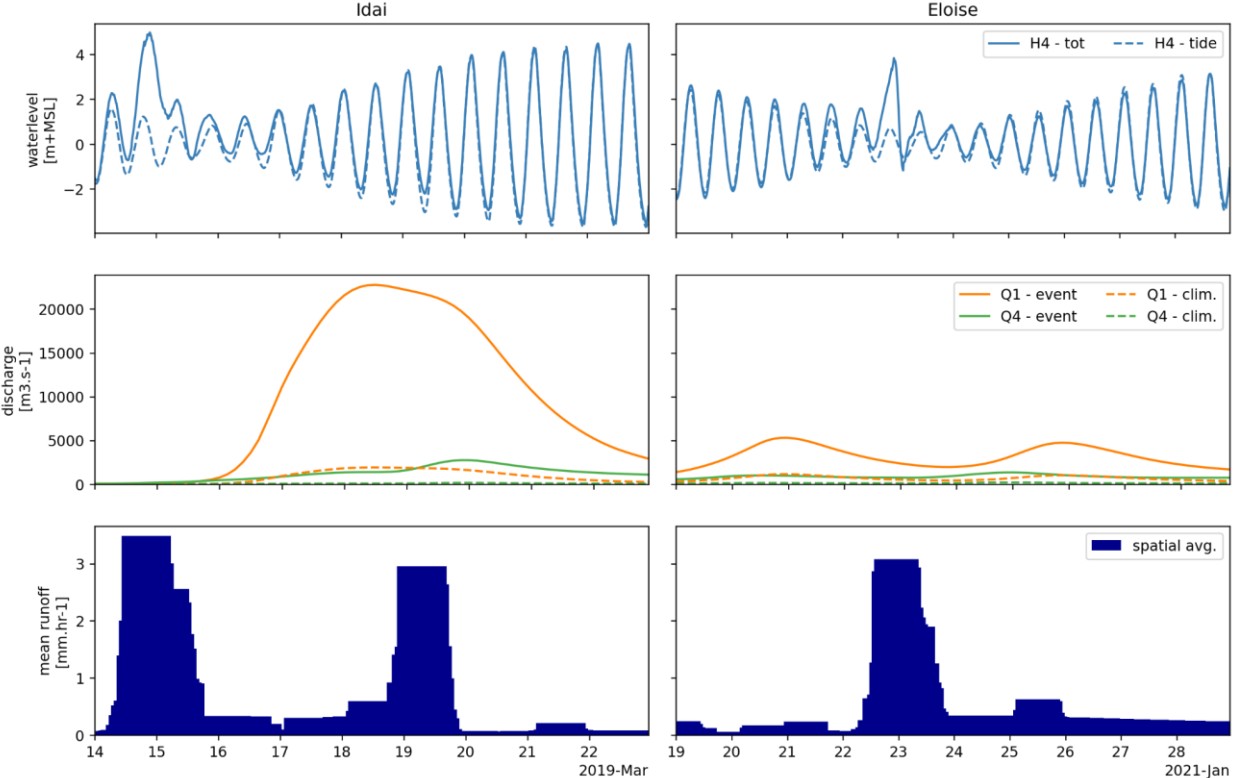

**Figure 3: SFINCS boundary conditions during cyclone Idai (left column) and cyclone Eloise (right column) for total sea level (top row); actual event discharge (center row); and spatial average runoff (bottom row). The full lines show the total water levels and event discharge, as used for the validation, see Section 3.3.1. The dashed lines show the tidal water level component only and normalized discharge to match the climatological mean, as used in the compound driver analysis, see Section 3.3.3. Only one coastal location (H4) and the two main rivers (Q1 - Buzi and Q4 - Pungwe) are shown to improve the readability of the plots. The labels correspond to the locations as shown in Figure 2B.**

## 3.3 Analysis of the model results

### 3.3.1 Validation against observed flood extent

As no quantitative stream flow or water level observation data are openly available for this location, we focus on a comparison between observed and simulated flood extent. Model skill is quantified based on three metrics that are commonly used to analyze flood models (Vousdoukas et al., 2016; Wing et al., 2021). The model skill is measured by the critical success index ($C$), which is the ratio of the area that is correctly simulated to be flooded ($Fsim \cap Fobs$) over the union of observed and simulated flooded areas ($Fsim \cup Fobs$), thereby accounting for both over- and underprediction, see eq. (1). The critical success index ranges from 0 (no match) to 1 (perfect match). The hit rate ($H$) is the ratio area that is correctly simulated to be flooded over the observed flood extent ($Fobs$) , see eq. (2). The hit rate ranges from 0 (none of the observed flood extent are flooded in the model) to 1 (the complete observed flood extent is flooded in the model). The false





alarm rate ($F$) is the ratio of the area which is wrongly simulated to be flooded ($Fsim/Fobs$) over the observed flood extent, see eq. (3). The false alarm rate ranges from 0 (no overprediction) to infinity (1 indicates equally sized areas of wrongly simulated and observed flooding).

$$C = \frac{Fsim \cap Fobs}{Fsim \cup Fobs} \qquad (1)$$

$$H = \frac{Fsim \cap Fobs}{Fobs} \qquad (2)$$

$$F = \frac{Fsim/Fobs}{Fobs} \qquad (3)$$

The high-resolution (10 m) observed flood extent is derived from Sentinel-1 synthetic aperture radar data and processed
using the Radar Produced Inundation Diary (RAPID) algorithm (Shen et al., 2019a). The algorithm has recently been applied to derive flood extent images based on the full stack of Sentinel-1 images over the contiguous United States with high accuracy (Yang et al., 2021). For this study the union of the flood extents from images intersecting with our study area on 20 March 2019 (during the tropical cyclone Idai flood event) and 25 January 2021 (during the tropical cyclone Eloise flood event) are used. The flood extents are reprojected to the SFINCS model grid and permanent water cells based on the RAPID
algorithm combined with the binary model river mask removed from the flood extent.

The simulated flood extent is based on the maximum flood depth at the same day as the observation. Cells within the permanent water mask or with a flood depth below a 15 cm threshold are masked out from the flood extent. A similar threshold is used in other flood studies (e.g. Wing et al., 2017). To benchmark the SFINCS simulation, the same comparison
is made based on CaMa-Flood flood depth simulations. The same postprocessing is applied to the CaMa-Flood flood depth maps, but after downscaling to a 3 arcsec grid (see Section 3.1.2) and reprojection to the SFINCS grid using nearest neighbor interpolation.

### 3.3.2 Sensitivity analysis

We perform a sensitivity analysis of the model skill by varying several model parameters and model forcing for both
historical events. A description of each model sensitivity run is provided in the table below.

**Table 2: Overview of model sensitivity runs.**

| Parameter | Description | Lower value | Upper value |
|---|---|---|---|
| 1. River depth | The river depth is varied by multiplying the coefficient $a$ in the power-law equation, see section 3.2.1 | 50% (a = 0.135) | 150% (a = 0.405) |





| 2. Land manning roughness | The spatially constant manning roughness value for land cells (flood plain manning roughness in CaMa-Flood) | 50% (0.05 sm$^{-1/3}$) | 150% (0.15 sm$^{-1/3}$) |
|---|---|---|---|
| 3. Coastal (H) forcing | Total water level forcing (tide, surge, and wave setup components) for both SFINCS and CaMa-Flood. | 80% | 120% |
| 4. Pluvial (P) and fluvial (Q) forcing | The ERA5 runoff forcing of CaMa-Flood and pluvial forcing of SFINCS. Based on the CaMa-Flood simulation, the fluvial forcing of SFINCS is also modified. | 80% | 120% |
| 5. Bifurcations | *CaMa-Flood only*. The number of elevation thresholds [0-10] at which a representative width for flow between floodplains of adjacent unit-catchments is described. Here, 10 by default. | 0 (no bifurcations) | 5 |
| 6. Resolution | *CaMa-Flood only*. The resolution at which unit-catchments are described. | N/A? | 200% (6 arcmin) |

### 3.3.3 Compound flood drivers

To examine the role of each driver and interactions between fluvial, pluvial, and coastal flood drivers on flood levels, we
perform a scenario analysis with the local SFINCS model where we vary the boundary conditions, see Table 3 for details.
During single driver events, the forcing of both other drivers is adjusted to non-extreme conditions, see dashed lines in
Figure 3. For the fluvial boundary condition, we normalize the event discharge to match the long-term mean discharge; for
the pluvial boundary we set the rainfall to zero; and for the coastal boundary we use the tidal signal of the event only. We
identify transition zones as areas where water levels in the compound scenario are at least 5 cm higher than in any of the
single driver scenarios, in line with earlier studies on compound flooding where thresholds vary between 0–20 cm (Bilskie
and Hagen, 2018; Gori et al., 2020b; Shen et al., 2019b). In addition, we identify the main flood driver based on the single
driver scenario that results in the largest water level.

**Table 3: Overview of model boundary conditions in compound and single driver scenarios.**

| Scenario | Fluvial boundary | Pluvial boundary | Coastal boundary |
|---|---|---|---|
| Compound | CaMa-Flood event discharge | ERA5 event runoff | GTSM event tide and surge + ERA5 waves |
| Fluvial (single) | CaMa-Flood event discharge | none | GTSM event tide levels |
| Pluvial (single) | CaMa-Flood event discharge scaled to match long-term mean | ERA5 event runoff | GTSM event tide levels |
| Coastal (single) | CaMa-Flood event discharge scaled to match long-term mean | none | GTSM event tide and surge + ERA5 waves |






## 4. Results and discussion

### 4.1 Model comparison

In this section we present a comparison of the skill of the global CaMa-Flood and local SFINCS models to simulate the flood extent of the historical flood events Idai and Eloise. Both models are forced with the same data and we used the same

manning roughness and river depth estimation for compatibility. In general, we simulate more widespread flooding during cyclone Idai compared to Eloise and with SFINCS compared to CaMa-Flood (Figure 4). The difference between both models in the Buzi floodplains is likely due to the limited connectivity between floodplains of neighboring cells in the CaMa-Flood model. This can be seen in the downscaled CaMa-Flood flood maps, which show unrealistic sudden local drops in flood depth at the interface of unit catchments during cyclone Idai (Figure 4A) and larger simulated water levels in the

Buzi in CaMa-Flood compared to SFINCS (Figure 5A/B). The difference around the Pungwe estuary is likely due to the response of both models to coastal boundary conditions. Water levels in the Pungwe estuary in CaMa-Flood are more attenuated and slower compared to SFINCS (Figure 5C/D) due to the lower resolution of the CaMa-Flood model. In addition, some small coastal areas at the estuary mouth which are flooded in SFNCS are not covered by the CaMa-Flood model. The differences around Beira, where no flooding is simulated by CaMa-Flood, can be attributed to the fact that

CaMa-Flood does not simulate direct coastal flooding, but only the effect of coastal forcing on riverine water levels and subsequent fluvial flooding. Finally, the difference on the hillslopes can be attributed to the fact that CaMa-Flood does not simulate direct pluvial flooding. While in SFINCS the runoff forcing (i.e. net precipitation) is added as source term to each grid cell, in CaMa-Flood it is directly added to the river component of each unit catchment. Furthermore, the drainage capacity in this area in SFINCS is likely underestimated due to the absence of small (sub-grid scale) streams in the model

topography which is limited by the model resolution.







**Figure 4: Simulated flood depths from CaMa-Flood (left panels) and SFINCS (right panels) for cyclone Idai (top panels) and cyclone Eloise (bottom panels). The diamonds indicate model observation points for which water level time series are extracted, see Figure 5.**






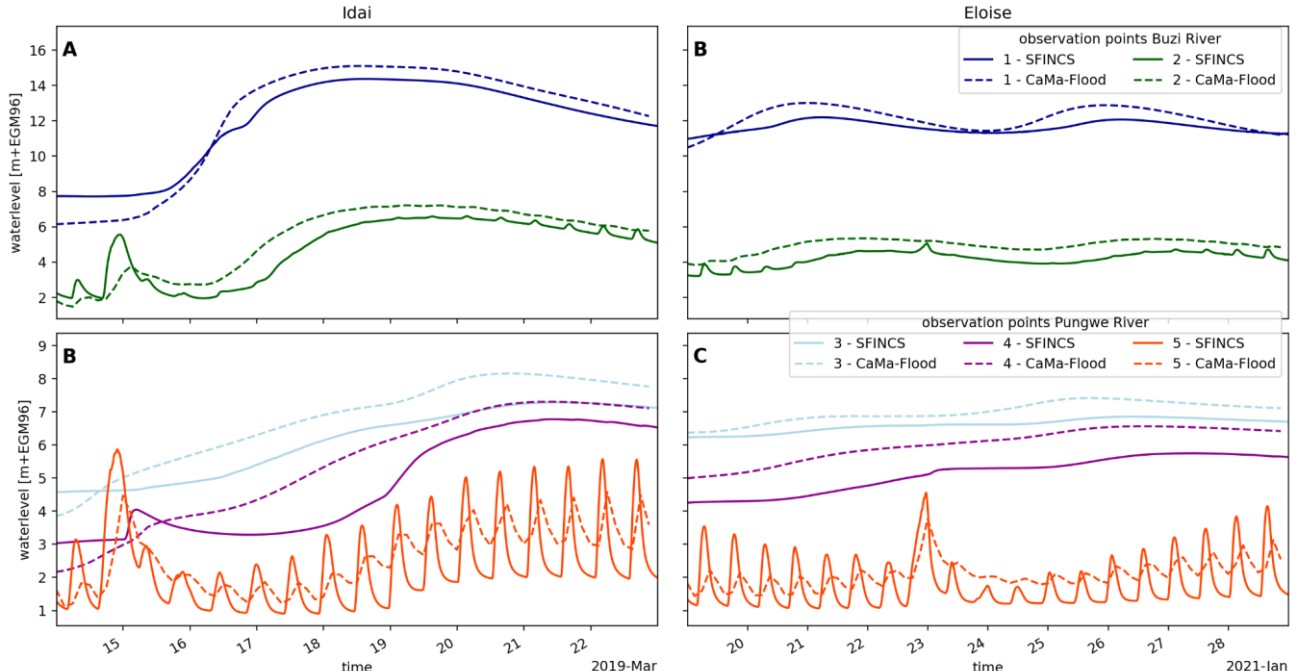

**Figure 5: Simulated time series of water levels during cyclone Idai (left) and cyclone Eloise (right) with SFINCS (full lines) and CaMa-Flood (dashed lines) for two locations in the Buzi (top) and three in the Pungwe (bottom). See diamonds in Figure 4 for the exact locations.**

For both events, we compare the simulated flood extents with observations from Sentinel-1 data. In terms of critical success index (C), SFINCS and CaMa-Flood perform similarly. The skill of both models is higher for the Idai flood event, where we find C = 0.72 for both the models (Figure 6A/C), while for the Eloise flood event we find C = 0.47 for CaMa-Flood and C = 0.44 for SFINCS (Figure 6B/D). Despite the similar skill, there are substantial differences in the simulated flood levels and flood extents between both models. The SFINCS simulations show larger flood extents compared to CaMa-Flood, resulting

in a higher hit ratio (H = 0.94 vs 0.83 during Idai and 0.87 vs 0.66 during Eloise) and a higher false alarm ratio (F = 0.24 vs 0.14 during Idai and 0.53 vs 0.37 during Eloise). The underestimation of CaMa-Flood is concentrated in the floodplains of the Buzi river and around and north of the city of Beira, see orange colors in Figure 6A/B. The overestimation in SFINCS is concentrated along the banks of the Pungwe river, the hillslopes in the north-easter corner of the model domain, and for Eloise also the floodplains south of the Buzi river, see red colors in Figure 6C/D.

To further investigate the model performance, we performed a sensitivity analysis on some of the most important parameters and forcing. In general, we find that the skill of both models is not very sensitive to the river depth (Table 4 - 1) and the manning roughness for land cells (Table 4 - 2). This is due to the extremeness of the fluvial driver, especially during cyclone Idai, during which the river conveyance capacity is small compared to the total discharge. Both models are also not sensitive to changes in the coastal forcing (Table 4 - 3) but are sensitive to changes in the pluvial and fluvial forcing, (Table 4 - 4).

This is due to the relatively large fluvial flood driver compared to the coastal flood driver during these two events and the





small fraction of the total flood area that is caused by direct coastal flooding (Section 4.2). For CaMa-Flood, we find that the model is very sensitive to the presence of bifurcation channels (Table 4 - 5) and resolution of the model (Table 4 - 6). With less bifurcation layers and a coarser model resolution the connectivity between floodplains reduces, resulting in a large decrease in model skill. Flow connectivity in the model has been shown to be important to correctly simulate inundation
dynamics in (coastal) floodplains (Bernhofen et al., 2018; Neal et al., 2012; Trigg et al., 2012). Multiple downstream connectivity, as implemented in the bifurcation scheme of CaMa-Flood, is crucial to adequately simulate floods in deltas (Ikeuchi et al., 2015; Mateo et al., 2017), which is underlined by the results in our study. However, we still find that the flow connectivity is underrepresented compared to the SFINCS mode as shown by the more widespread (fluvial) flooding with SFINCS.

The skill of both models is in line with other flood studies using global models. Global flood models showed C = 0.45–0.70 in comparison with MODIS imagery of three flood events over the African continent (Bernhofen et al., 2018) and C = 0.43–0.65 in comparison with various reference flood maps in Germany and the UK (Alfieri et al., 2014). A Lisflood-FP model build with LFPtools was found to have C = 0.63 for a flood event in the river Severn (Sosa et al., 2020). For local fluvial inundation models that are calibrated against flood extent imagery typical CSI of 0.7–0.9 can be reached, depending on the
quality of the flood extent imagery (Altenau et al., 2017; Di Baldassarre et al., 2009; Horritt and Bates, 2002; Stephens and Bates, 2015; Wood et al., 2016). Our results also demonstrate that a commonly used metric to evaluate flood models such as the critical success index can mask large differences between model results and should be evaluated together with the false alarm and hit ratios and inspection of the geographical patterns and differences. An additional comparison with flood levels, if available, would allow for an even more comprehensive validation (Stephens and Bates, 2015; Wing et al., 2021).






**Figure 6: Comparison of simulated flood extents from CaMa-Flood (top panels) and SFINCS (bottom panels) for cyclone Idai (left panels) and cyclone Eloise (right panels), evaluated based on critical success index (C), hit-rate (H) and false alarm ratio (F) as shown in the top right of each panel.**





**Table 4: Sensitivity analysis of modeled flood extent with CaMa-Flood (CMF) and SFINCS (SF) in comparison with observations to river depth, manning roughness, coastal driver (H forcing), pluvial and fluvial drivers (P & Q forcing), bifurcations and spatial resolution. The flood extent is evaluated in terms of critical success index (C), hit-rate (H) and false alarm ratio (F). For scenarios 1-6 the difference in skill relative to the base scenario is shown, the largest absolute differences per column are highlighted.**

| | Idai | | | | | | Eloise | | | | | |
|---|---|---|---|---|---|---|---|---|---|---|---|---|
| | C | | F | | H | | C | | F | | H | |
| | CMF | SF | CMF | SF | CMF | SF | CMF | SF | CMF | SF | CMF | SF |
| **0. default** | **0.72** | **0.72** | **0.14** | **0.24** | **0.83** | **0.94** | **0.47** | **0.44** | **0.37** | **0.53** | **0.66** | **0.87** |
| 1a. river depth: 50% | 0.00 | 0.00 | 0.01 | 0.00 | 0.01 | 0.00 | 0.00 | 0.00 | 0.03 | 0.01 | 0.04 | 0.01 |
| 1b. river depth: 150% | 0.00 | 0.00 | -0.01 | -0.01 | -0.01 | 0.00 | -0.01 | 0.01 | -0.03 | -0.01 | -0.05 | -0.01 |
| 2a. land manning: 50% | -0.05 | *-0.01* | -0.01 | -0.01 | -0.07 | *-0.03* | -0.02 | 00.0 | -0.02 | -0.03 | -0.06 | *-0.07* |
| 2b. land manning: 150% | 0.00 | *-0.01* | 0.00 | 0.01 | 0.00 | 0.00 | 0.02 | -0.01 | 0.01 | 0.02 | 0.05 | 0.03 |
| 3a. H forcing: 80% | 0.00 | 0.00 | 0.00 | 0.00 | 0.00 | 0.00 | 0.00 | 0.00 | 0.00 | 0.00 | 0.00 | 0.00 |
| 3b. Hl forcing: 120% | 0.00 | 0.00 | 0.00 | 0.01 | 0.00 | 0.00 | 0.00 | 0.00 | 0.00 | 0.00 | 0.00 | 0.00 |
| 4a. P & Q forcing: 80% | -0.03 | *0.01* | -0.02 | -0.03 | -0.05 | *-0.03* | -0.02 | *0.02* | *-0.04* | *-0.04* | -0.07 | -0.05 |
| 4b. P & Q forcing: 120% | 0.01 | *-0.01* | *0.02* | 0.02 | 0.04 | 0.01 | 0.01 | -0.01 | 0.03 | 0.02 | 0.05 | 0.03 |
| 5a. bifurcations: 50% | -0.03 | N/A | *0.02* | N/A | -0.02 | N/A | 0.01 | N/A | -0.01 | N/A | 0.00 | N/A |
| 5b. bifurcations: 0% (off) | *-0.26* | N/A | *0.02* | N/A | *-0.32* | N/A | -0.09 | N/A | 0.04 | N/A | -0.14 | N/A |
| 6. spatial res: 200% | -0.06 | N/A | -0.01 | N/A | -0.08 | N/A | *-0.13* | N/A | 0.03 | N/A | *-0.21* | N/A |

## 4.2 Potential application: compound flood drivers

To showcase a possible application of the compound flood model framework and the added value over the global model, we examine the role of each driver and interactions between flood drivers for both events (Figure 7). The difference in maximum water levels between the compound scenario and the single flood driver scenario that results in the largest flood depth (i.e. the dominant flood driver) is shown in the top panels. The bottom panels show the dominant flood driver with green (pluvial), purple (fluvial) or orange (coastal) colors, which are darker for transition zones, where interactions between

drivers amplify the total water level (i.e. show a positive difference in the top panels larger than 5 cm). For most of the model domain, the dominant flood driver during both events is fluvial, especially around the Buzi river and the upstream part of the Pungwe river. The coastal flood driver is dominant in the most downstream ends of both estuaries and in small coastal areas around Beira. Pluvial drivers are dominant on the hill slopes in the north east corner of the model domain, but mainly add to fluvial and coastal driven flooding. When we compare both events, we find that, for Eloise, the extent where the





coastal driver is dominant as well as the amplification of water levels in the transition zones are larger compared to the Idai event. This can be explained by the difference in fluvial flood magnitude and the timing between the peaks of fluvial and coastal drivers. During the Idai event sea water levels peaked around March 15, followed by a discharge peak at the Buzi river three days later and the Pungwe river 5 days later (Figure 3 left panels), causing little interaction between the fluvial and coastal drivers. In the Pungwe estuary we even find a small decrease in the compound scenario compared to the coastal

and fluvial single driver scenarios, which is due to a small (< 0.1 m) negative non-tidal residual at high tide. During the Eloise event a first discharge peak at the Buzi river occurred two days before the coastal water level peak on January 23, followed by a small peak in the Pungwe river 1.5 days later and another large peak in the Buzi river 3 days later (Figure 3 right panels), causing a large (> 0.2 m) amplification of the water levels in both rivers. We also investigate the sensitivity of the transition zone for river and estuarine bathymetry. For the sensitivity simulation with deeper bathymetry (simulation 2b

in Table 2 and 4) the transition zone in the Pungwe estuary extends a bit further inland (Figure A2). While these changes are relatively small, the accuracy of the river and estuarine bathymetry is clearly important to accurately determine the transition zone.

Compared to earlier research that focused on interactions between coastal and pluvial drivers (Bilskie and Hagen, 2018; Gori

et al., 2020b), we derive transition zones based on three drivers and distinguish between the fluvial and pluvial drivers. In line with the aforementioned studies, our results also demonstrate that a single map with discrete transition zones for a specific region does not exist. A comprehensive overview of flood transition zones could be derived based on the occurrence of compounding effects across a large range of plausible events. The relative timing between peaks of different flood drivers as well as their magnitude has a large effect on the locations and area of transition zones. This is also underlined by a recent

study that found that compound flood levels are sensitive to the timing between flood peaks, especially for events and locations where the duration of discharge peaks is relatively short (Harrison et al., 2021).





**Figure 7: Compound flood dynamics during the Idai flood event (left panels) and the Eloise flood event (right panels) illustrated by the difference between water levels from the compound flood scenario and the maximum of all single driver scenarios (top panels); and the main flood driver based on the single driver scenario with the maximum water level (bottom panels). The main driver is indicated with light colors where the water level results for a single flood driver and dark colors where it results from more than one flood driver, also referred to as transition zone.**



### 4.3 Limitations and recommendations

While the model framework based on global open-source datasets comes with large benefits in terms of global applicability,
the accuracy of the input data is an important consideration. River and estuarine bathymetry are a relatively large source of
uncertainty in the current model setup. As bathymetry cannot be directly observed remotely, it needs to be approximated in
data-scarce areas where no local measurements are available. This approximation can have a large effect on the result of
(compound) inundation simulations (Harrison et al., 2021; Neal et al., 2012; Sampson et al., 2015). Better methods to
estimate bathymetry, such as the recently published gradual varying flow theory based method (Neal et al., 2021; Garambois
and Monnier, 2015) and new data such as expected from the surface water and ocean topography (SWOT) mission
(Andreadis et al., 2020), are expected to be useful to further reduce this uncertainty. For streams smaller than the model
resolution, a subgrid schematization could further improve the model (Neal et al., 2012; Volp et al., 2013). A subgrid
schematization has recently been implemented in SFINCS (Leijnse et al., 2020) and has been applied by Röbke et al. (2021)
for tsunami flood modeling. Furthermore, uncertainties in the global DEM (Hawker et al., 2018a; Hinkel et al., 2021) and the
absence of information on flood defense structures in many areas (Scussolini et al., 2016; Wing et al., 2019) may have large
implications for the accuracy of the flood simulations. The framework is set up such that datasets can easily be replaced by
better (local) datasets which also facilitates the update of new datasets in future model versions, such as the recently
published FABDEM, which is a for vegetation and building bias corrected version of the 30m resolution Copernicus DEM
(Hawker et al., 2022).

Forcing data are an important source of uncertainty for flood modeling in ungauged areas (Hoch et al., 2019; Wing et al.,
2020). For the selected case study ground observations are very scarce and comparisons with simulated discharge and total
sea levels are conducted qualitatively, see Section 3.1. We recommend investigating whether remote sensing, e.g. satellite
laser or radar altimetry data, can be used to validate extreme inland and nearshore water levels (Andreadis et al., 2020;
O'Loughlin et al., 2016; Urban et al., 2008). The hydrodynamic model was validated based on flood extent for two events as
observed by the Sentinel-1 satellites. However, flood extents based on SAR data are known to have limitations in observing
obstructed flooding such as in wetland or urban areas (Yang et al., 2021). To further increase the credibility of the model it
should be validated against a larger set of flood events, for instance using the recently published Global Flood Database
based on MODIS data (Tellman et al., 2021) or the RAPID sentinel-1 database over the continental United States (Yang et
al., 2021).

Furthermore, both events are characterized by a large significant height of wind waves (wave setup component amounts to
24.4% for Idai and 16.3% for Eloise of total water levels), indicating that wave setup could not be ignored. In this study we
used a simple approach to estimate wave setup justified by our aim to make the framework globally applicable. The wave
setup component could potentially be improved using alternative methods which use additional wave and morphological
parameters (e.g. Stockdon et al., 2006), possibly in combination with a recently published dataset on nearshore slopes



(Athanasiou et al., 2019). The large computational costs of wave models due to the high required spatiotemporal resolution still prohibits their direct application on large spatial scales (Hinkel et al., 2021). However, these models can still be leveraged for large scale flood risk applications by developing large synthetic databases of model results for many different plausible cross sections under varying forcing conditions (van Zelst et al., 2021; Pearson et al., 2017).

## 5. Conclusions

In this study we present an automated framework to model compound flooding anywhere on the globe in a reproducible and transparent manner; we evaluate its suitability and use it to identify compound flood drivers. The framework is comprised of the local 2D flood model SFINCS, set up based on global datasets and forced by global models at its boundaries. For two historical compound flood events in the Sofala province of Mozambique, we compared the skill of the local flood model with the global quasi 2D CaMa-Flood model. The validation against flood extents from satellites shows a good model

performance. While both models show similar skill in terms of critical success index, large differences exist in the simulated flood maps. Firstly, the local model can accommodate for direct coastal and pluvial flooding as well as interactions between coastal, pluvial, and fluvial drivers, thereby providing a more comprehensive description of flooding in coastal deltas than the global model. Therefore, the local model results have a higher hit rate of observed flood extents. Secondly, while the multiple downstream connectivity (or bifurcation) scheme largely improves the results of the global model, the floodplain

connectivity is still limited compared to the local model, resulting in higher flood levels and smaller flood extents. Thirdly, pluvial flooding is likely overestimated in the local model as small streams are not represented in the model underestimating the drainage capacity. We hypothesize that this will improve with the recently implemented subgrid schematization in SFINCS in combination with higher resolution DEMs. Finally, we show that the local model can be used to analyze the effect of interactions between flood drivers, here for the first time presented with joint fluvial, pluvial, and coastal flood

drivers. We find that the transition zones between flood drivers vary significantly between flood events due to differences in the relative timing between and magnitude of each driver. As the identification of these zones is important to understand flood preparedness and response, their identification should therefore be based on a large number of plausible flood events. We also reiterate the importance of observed water levels for a more comprehensive comparison of flood simulations.

The automated model setup is available through the open–source python package HydroMT-SFINCS and allows for a fast and reproducible setup of compound flood hazard models. With sufficient computational resources, the framework therefore has the potential to be scaled up to large spatial scales by setting up many local high-resolution models in river and coastal floodplains but could also rapidly be employed for disaster response.



## Data and code availability

The scripts and data used to setup the experiments in this study are available from Zenodo at https://doi.org/10.5281/zenodo.6413642/

## Author contributions

DE, HI, and PJW conceived the idea for this study; DE designed and executed the experiments with important inputs from PJW, HC and AC; JD & SM provided the necessary GTSM simulations; DY provided the CaMa-Flood model; DE, TL & HC developed the HydroMT-SFINCS plugin which is at the basis of the experiment; DE wrote the manuscript with input from all authors.

## Competing interests

The authors declare that they have no conflict of interest.

## Acknowledgements

We would like to thank Qing Yang RAPID based Sentinel-1 SAR data. The research leading to these results received funding from the Netherlands Organization for Scientific Research (NWO) in the form of a VIDI grant (Grant No. 016.161.324) and internal SO research funding by Deltares. PJW received funding from the European Union's Horizon 2020 research and innovation programme under grant agreement No 101003276 (MYRIAD-EU). SM received funding from the research programme MOSAIC with project number ASDI.2018.036, which is financed by NWO. Contributions of DY and HI are supported by JSPS KAKENHI 21H05002





**Appendix A - supporting figures**

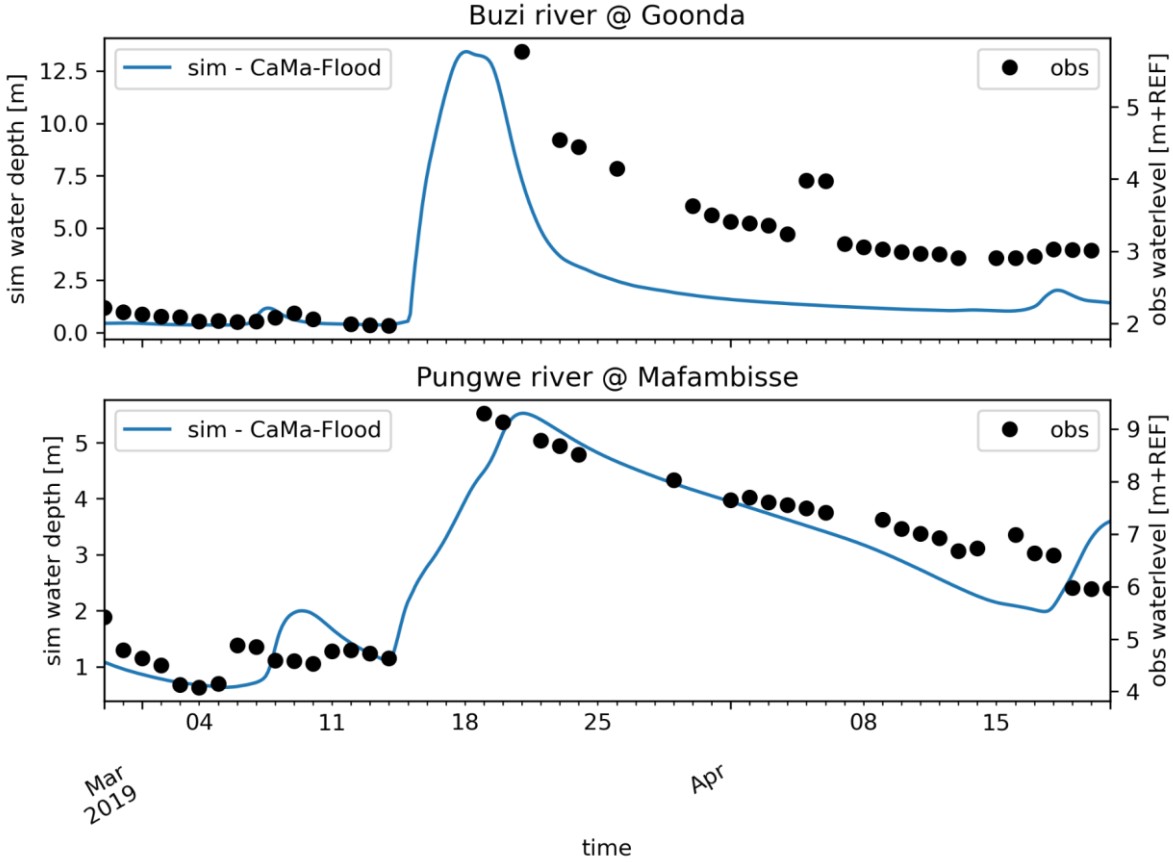

**Figure A1: Comparison of observed water levels and simulated water depths during tropical cyclone Idai in the Buzi (top) and**
570 **Pungwe (bottom) river. Note that the comparison is based on approximate locations as the precise locations could not be retrieved.**
**Furthermore, as the vertical datum of the observations is unknown these are plotted on a second y-axis.**





**Figure A2: Sensitivity analysis of compound flood dynamics simulation based on 150% river depth during the Idai flood event (left) and the Eloise flood event (right) illustrated by the difference between water levels from the compound flood scenario and the**
575 **maximum of all single driver scenarios (top panels); and the main flood driver based on the single driver scenario with the maximum water level (bottom panels). The main driver is indicated with light colors where the water level results for a single flood driver and dark colors where it results from more than one flood driver, also referred to as transition zone.**





## Appendix B - HydroMT configuration

**Table B1: Example HydroMT-SFINCS configuration file used to setup the SFINCS model schematization (see Section 3.2.1). Each section corresponds to a step in the automatic model building process. Options ending with _fn (filename) correspond to data from the data catalog, see Table B2.**

```
[setup_config]
alpha = 0.5
qinf = 0.0
dtout = 86400

[setup_topobathy]
topobathy_fn = merit_hydro
crs = utm

[setup_river_hydrography]
hydrography_fn = merit_hydro
adjust_dem = True
outlets=edge

[setup_river_bathymetry]
river_geom_fn = rivers_lin2019_v1
river_mask_fn = grwl_mask
rivwth_method = mask
rivdph_method = gvf
river_upa = 100
constrain_estuary = True
rivbank = True

[setup_mask]
drop_area = 1000
reset_mask = True

[setup_river_inflow]
river_upa = 500
river_len = 10e3

[setup_bounds]
btype = waterlevel
mask_fn = osm_coastlines
mask_buffer = 200
```




```
[setup_river_outflow]
river_upa=10
outflow_width=1e3

[setup_gauges]
gauges_fn=obs_locs.geojson
```

**Table B2: Data catalog (yaml) file used to set up the SFINCS model schematization (see Section 3.2.1). Each entry corresponds to a dataset and contains information about how to read it and which preprocessing steps (such as renaming) are required.**

```
grwl_mask:
  data_type: RasterDataset
  driver: raster
  meta:
    paper_doi: 10.1126/science.aat0636
    paper_ref: Allen and Pavelsky (2018)
    source_license: CC BY 4.0
    source_url: https://doi.org/10.5281/zenodo.1297434
    source_version: 1.01
  nodata: 0
  path: grwl_mask.tif
mdt_cnes_cls18:
  crs: 4326
  data_type: RasterDataset
  driver: raster
  meta:
    paper_doi: 10.5194/os-17-789-2021
    paper_ref: Mulet et al (2021)
    source_url: https://www.aviso.altimetry.fr/en/data/products/auxilia[..]
    source_version: 18
    unit: m+GOCO05S
  path: mdt_cnes_cls18.tif
merit_hydro:
  crs: 4326
  data_type: RasterDataset
  driver: raster
  meta:
    paper_doi: 10.1029/2019WR024873
    paper_ref: Yamazaki et al. (2019)
    source_license: CC-BY-NC 4.0 or ODbL 1.0
    source_url: http://hydro.iis.u-tokyo.ac.jp/~yamadai/MERIT_Hydro
```





```
    source_version: 1.0
  path: merit_hydro\{variable}.tif
osm_coastlines:
  crs: 4326
  data_type: GeoDataFrame
  driver: vector
  meta:
    source_author: OpenStreetMap
    source_info: OpenStreetMap coastlines water polygons, last updated 2020-01-09T05:29
    source_license: ODbL
    source_url: https://osmdata.openstreetmap.de/data/coastlines.html
    source_version: 1.0
  path: osm_coastlines.gpkg
rivers_lin2019_v1:
  data_type: GeoDataFrame
  driver: vector
  meta:
    paper_doi: 10.5281/zenodo.3552776
    paper_ref: Lin et al. (2019)
    source_license: CC-BY-NC 4.0
    source_url: https://zenodo.org/record/3552776#.YVbOrppByUk
    source_version: 1
  path: rivers_lin2019_v1.gpkg
  rename:
    width_m: rivwth
    Q2: qbankfull
```

585





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
