# Peer review of "A globally-applicable framework for compound flood hazard modeling"

_EGUsphere, 2022_

## Referee Comment (RC1)

Review of: *A globally-applicable framework for compound flood hazard modeling,* **Eilander et al.**

**General Comments:**

This article presents a new flood modelling framework, focusing on compound flood hazard from fluvial, pluvial and coastal flooding. It is well written and provides appropriate detail around the methodology. The model, which uses global inputs to build a local scale flood model will be a valuable tool to investigate compound flooding.

The models appear to have skill on par with other models built on global data, although additional case studies and/or validations would be nice to see how the framework performs in a range of different situations.

I would recommend the article for publication with minor modifications:

**Specific Comments:**

1.  As above, case studies for a different location would strengthen the validation.

2.  Validation observations:

    The observations used in the validation are based on single snapshots for each event.

    ○  Firstly it would be useful to add some context at where these snapshots fit into the event (how close are they to the fluvial/ coastal flooding peaks for example).

    ○  Similarly, can you comment on how the timing of the observations influences the validation?  E.g., Would the results in Table 4 -3 showing reduced sensitivity to coastal drivers be connected to the time of observations being before/after the coastal flooding had receded?

    ○  Are there any alternative observations of the flood hazard for the same events you could use (e.g. extents from Copernicus EMS)?

3.  It seems like there are only river inputs at the boundaries of the domain, is this correct?

    Does that mean in the coupled framework, the pluvial rain/runoff on-grid implementation is relied on to convey water in tributaries completely within the domain?

    Is there likely to be an underestimation of river flows due to this (for example if this framework is used to simulate larger domains)?

**Technical comments/ corrections:**

1. Abstract: You refer to 'local scale' models (L13) to refer to models built with high quality local information, but then refer to the SFINCS model as a 'local model' in L22 and L24. Could you reword this to make it clear these are referring to different things?

2. L17: 'loosely coupled' is not clear, consider rewording.

3. L135: This is a little confusing. It would be useful to know that the total water height consists of two components (L138-140) first. Also that the ERA5 data comes from the ECMWF Ocean Wave Model.

4. L144: It's not too clear where the gauge locations are here.

5. Table 1: Are you using MERIT DEM or the MERIT-hydro hydrologically adjusted elevations?

6. L255: should this final sentence have a separate bullet point?

7. L302: is the 'gridded discharge dataset' referring to CaMa-flood?

8. L307: add: maximum relative error in upstream area

9. Figure 6: Panels are mislabelled

10. L552: The zenodo link doesn't work (but does if the final slash is removed).

---

## Author Comment (AC1)

**Reply on RC1**

**General Comments:**

This article presents a new flood modelling framework, focusing on compound flood hazard from fluvial, pluvial and coastal flooding. It is well written and provides appropriate detail around the methodology. The model, which uses global inputs to build a local scale flood model will be a valuable tool to investigate compound flooding.

The models appear to have skill on par with other models built on global data, although additional case studies and/or validations would be nice to see how the framework performs in a range of different situations. I would recommend the article for publication with minor modifications:

*We would like to thank the reviewer for the thorough review and comments, which we believe have led to an improvement in the manuscript. We are pleased to read that the reviewer considers the manuscript to be well written and the presented framework a valuable tool for compound flood analysis. Based on the suggestions of both reviewers we have made some changes to the manuscript.*

*To improve the robustness of the analysis we have also extended the validation dataset with two more flood extent images, one for each event. To obtain this extended dataset in a consistent manner, we have processed the Sentinel-1 images with a slightly different algorithm as this algorithm was available to our team (see section 3.3.1). The new algorithm provides very similar results compared to the RAPID product used in the original submission. Furthermore, we have made some small improvements to the model setup by including more small rivers as boundary conditions to the SFINCS model and focus the analysis on the areas connected to the Buzi and Pungwe floodplains.*

*Our response to the specific comments can be found in the paragraphs below.*

**Specific Comments:**

1. As above, case studies for a different location would strengthen the validation.

*For this manuscript we have deliberately selected only one case study which allows us to provide more context on the studied events and to demonstrate how the framework can be used to analyze compound events. We have clarified the goal of the paper at the end of the introduction. However, we agree that additional validation would be useful in further research, as mentioned in the discussion.*

*L70: The goal of this study is to present the framework, to test its ability to simulate compound floods in data-sparse coastal deltas, and to demonstrate how it can be used for compound flood analysis*

*L518: To further increase the credibility of the model it should be validated against a larger set of flood events, for instance using the recently published Global Flood Database based on MODIS data (Tellman et al., 2021) or the RAPID sentinel-1 database over the continental United States (Yang et al., 2021).*

2. Validation observations: The observations used in the validation are based on single snapshots for each event.

- Firstly it would be useful to add some context at where these snapshots fit into the event (how close are they to the fluvial/ coastal flooding peaks for example).
- Similarly, can you comment on how the timing of the observations influences the validation? E.g., Would the results in Table 4 -3 showing reduced sensitivity to coastal drivers be connected to the time of observations being before/after the coastal flooding had receded?
- Are there any alternative observations of the flood hazard for the same events you could use (e.g. extents from Copernicus EMS)?

Thanks for these suggestions. We have expanded the analysis by processing additional Sentinel-1 observations using an unsupervised histogram-based surface water mapping algorithm (Markert et al., 2020). We did not use additional images from the Copernicus EMS as these only cover a part of our area of interest and are mostly based on the same Sentinel-1 images. For each event we now have two observations on subsequent days. We compare the individual observed flood extents with the maximum simulated extent from the same day and the maximum observed extent per event with the maximum simulated extent during all days with observations. The observations during Idai are around the flood peak and for Eloise just before the flood peak. We describe the algorithm and the temporal context of the observations in section 3.3.1 (L339 & L452)

The sensitivity analysis is based on the maximum extents per event. We added a sentence in section 4.1 (L424) to clarify that the sensitivity to the drivers might be different if compared for a snapshot after the surge peak.

*L339: High-resolution (10 m) flood extent data are derived from Sentinel-1 Synthetic Aperture Radar (SAR) images. We use VV-polarized ground range detected level data, provided by Google Earth Engine (GEE), which has undergone geometric terrain correction and provides radar backscatter in decibel (dB) units. These data are processed using the GEE with an unsupervised histogram-based surface water mapping algorithm that consists of three steps (Markert et al., 2020). First, noise is reduced using the Refined Lee speckle filter (Lee, 1981). Second, a threshold to distinguish water and dry cells is detected using the Edge Otsu thresholding algorithm (Donchyts et al., 2016). Third, cells*

*with a relative elevation of more than 50 m above the nearest stream are excluded from the water class to avoid false positives. We process each image individually and combine flood extents from ascending and descending orbits during the same day. In total we obtain flood extents for four days based on eight images: on the 19 and 20 March 2019 for Tropical Cyclone Idai which is around the peak of the flood event, and on 25 and 26 January 2021 for Tropical Cyclone Eloise which is just before the peak of the flood event. Finally, the flood extents are reprojected to the SFINCS model grid.*

*L352: We compare the individual observed flood extents with the maximum simulated extent from the same day and the maximum observed extent per event with the maximum simulated extent during all days with observations.*

*L424: The skill is likely more sensitive to the coastal water level forcing if assessed for a snapshot around the surge peaks of both events instead of the multi-day maximum flood extent.*

3. It seems like there are only river inputs at the boundaries of the domain, is this correct? Does that mean in the coupled framework, the pluvial rain/runoff on-grid implementation is relied on to convey water in tributaries completely within the domain? Is there likely to be an underestimation of river flows due to this (for example if this framework is used to simulate larger domains)?

For this case study we indeed only applied discharge boundaries at the boundaries of the model domain where a river enters the model domain, where rivers are defined based on a minimal contribution area of 100 km2. Within the domain we apply the ERA5 runoff (i.e. net rainfall) directly to all the SFINCS grid cells and thus rely on SFINCS to convey the water. To prevent under/overestimating river flow in large domain models it is important to correctly schematize the conveyance capacity of the rivers and smaller streams (see also section 4.3).

**Technical comments/ corrections:**

1. Abstract: You refer to 'local scale' models (L13) to refer to models built with high quality local information, but then refer to the SFINCS model as a 'local model' in L22 and L24. Could you reword this to make it clear these are referring to different things?

Good suggestion. We now refer to the SFINCS model within the presented framework as the "globally-applicable model" throughout the manuscript..

2. L17: 'loosely coupled' is not clear, consider rewording.

We have removed loosely in the abstract and introduction, but explain it in more detail in the methods:

*L282: SFINCS is forced based on output from global models, which is automatically transformed to the input data format that SFINCS requires. This is also referred to as a loose coupling between models (Santiago-Collazo et al., 2019).*

3. L135: This is a little confusing. It would be useful to know that the total water height consists of two components (L138-140) first. Also that the ERA5 data comes from the ECMWF Ocean Wave Model.

We have clarified that the ERA5 significant wave height data originates from the ECMWF OCean Wave Model.

*L129: Hourly time series of significant height of wind waves (Hs) are extracted at GTSM output locations from the 30 arcmin ERA5 dataset and based on the ECMWF Ocean Wave Model (Bidlot, 2012; Hersbach et al., 2020).*

4. L144: It's not too clear where the gauge locations are here.

We have clarified the location (port of Beira) in the text.

*L149: In comparison with the tidal constituents of International Hydrographic Organization (IHO) station at the Port of Beira as retrieved using the Delft Dashboard (van Ormondt et al., 2020), […]*

5. Table 1: Are you using MERIT DEM or the MERIT-hydro hydrologically adjusted elevations?

We use the MERIT-hydro dataset as these data provide a better estimate of river elevation, which we use to estimate the river bathymetry (see section 3.2.1).

6. L255: should this final sentence have a separate bullet point?

Yes, we have changed the text accordingly.

7. L302: is the 'gridded discharge dataset' referring to CaMa-flood?

Yes, that's correct. For step 7-9 we first discuss the general implementation, followed by the specific data and choices made for the case study. In L304 we explain that for this case study we use discharge from the gridded CaMa-Flood output.

8. L307: add: maximum relative error in upstream area

Good suggestion. We have changed the text accordingly.

*L304: [..] based on a maximum relative error of 5% or absolute error of 100 km2 in upstream area.*

9. Figure 6: Panels are mislabelled

In this case the caption of Figure 6 was incorrect (not the labels). We have changed the caption to match the labels.

10. L552: The zenodo link doesn't work (but does if the final slash is removed).

We have fixed the URL.

**References**

Markert, K. N., Markert, A. M., Mayer, T., Nauman, C., Haag, A., Poortinga, A., Bhandari, B., Thwal, N. S., Kunlamai, T., Chishtie, F., Kwant, M., Phongsapan, K., Clinton, N., Towashiraporn, P., and Saah, D.: Comparing Sentinel-1 Surface Water Mapping Algorithms and Radiometric Terrain Correction Processing in Southeast Asia Utilizing Google Earth Engine, Remote Sensing, 12, 2469, https://doi.org/10.3390/rs12152469, 2020.

---

## Author Comment (AC2)

**Reply on RC2**

The article presents a methodology for local-scale compound flood modeling using global input datasets and a newly developed hydrodynamic model (SFINCS). The framework is applied to a coastal catchment in Mozambique that recently experienced flooding from two tropical cyclones (Idai and Eloise). The proposed methodology for developing local-scale models based on global datasets/inputs is very interesting, and I believe the paper could be an important contribution to the compound modeling literature. However, I believe some more analysis/discussion on the model validation is needed before this work can be published. Therefore, I recommend a moderate revision.

My main concern/issue with the paper is that the results presented in section 4.1 are not compelling. It seems like the local-scale model and the global CaMa model perform similarly well for the two historical cases, which calls into question why someone should go through the trouble of setting up a high-resolution local model if similar accuracy can be achieved with an existing global model. To be clear, I believe there is a lot of value in using a high-resolution local model for flood hazard analysis, I just don't think the results presented in section 4.1 do a good job of showing the additional benefit. Can any additional validation data, performance metrics, discussion, etc. be added to this section to show more clearly the benefit of using the SFINCS model? The ability to efficiently set up and run local-scale compound flood models for any catchment across the globe is really promising, but we need more confidence that the local-scale model will provide higher accuracy compared to existing global models.

We would like to thank the reviewer for the thorough review and comments, which we believe have led to an improvement in the manuscript. We are pleased to read that the reviewer believes our manuscript is an important contribution to the compound modeling literature. Based on the suggestions of both reviewers we have made several changes to the manuscript. Our response to the specific comments can be found in the paragraphs below.

Most importantly we have tried to highlight the benefits of the globally-applicable framework better in the abstract and conclusions, see updated text below. We improved the validation with two more flood extent images, one for each event. Furthermore, we have made some small improvements to the model setup by including more small rivers as boundary conditions to the SFINCS model and focus the analysis on the areas connected to the Buzi and Pungwe floodplains.

*L18: To test the framework, we simulate two historical compound flood events, Tropical Cyclones Idai and Eloise in the Sofala province of Mozambique, and compare the simulated flood extents to satellite-derived extents at multiple days for both events. Compared to the global CaMa-Flood model, the globally-applicable model generally performs better in terms of the critical success index (-0.01 – 0.09) and hit rate (0.11 – 0.22), but lower in terms of false alarm ratio (0.04 – 0.14). Furthermore, the simulated flood depth maps are more realistic due to better floodplain connectivity and provide a more comprehensive picture as direct coastal and pluvial flooding are simulated.*

**I have some other specific comments below:**

**3.1.1** I wonder if an ocean model with 2.5 km coastal resolution can adequately capture peak storm tides from TCs, which tend to produce extreme storm surges over relatively small geographic areas (cite).

We agree with the reviewer that the resolution of the hydrodynamic model is important to correctly capture storm surge peaks as local bathymetry could have significant effects on the surge. However, Dullaart et al. (2020) have shown that the Global Tide and Surge Model (GTSM) forced with ERA5 was able to simulate the storm surge peak for five TC events with a bias ranging from -22 to -2 cm. Furthermore, for that case of Beira which is located on a wide continental shelf (~140 km) a 2.5km grid resolution is deemed sufficient to capture the storm. We argue that this level of uncertainty is adequate for a globally-applicable model framework considering the bias in available global elevation datasets.

**Lines 142-149:** I was confused here as the sources of the storm surge, wind setup, and tide heights were not clear. The 5.0 m max water level (and 3.8 m for Eloise) reported here is based on what? Gauge, high water marks, reports, models? What is the max water level predicted by the author's global model? I see 4.0 m as the max surge estimate, but what about the total modeled water level? Also, how is the "operational forecast" generated? Is this another global model that the authors compared with? In general I think this paragraph needs to be re-written to be clear about how their model results compare with the results from other models or other sources.

We have rewritten the paragraph to be more clear about the data sources.

*L136: Due to a lack of observations from coastal water level gauges, a quantitative validation of the simulated water levels is not possible, but some general observations about the simulated data can be made. The maximum simulated water levels in GTSM during both events (5.0 m+MSL during Idai; 3.8 m+MSL during Eloise) occurred close to neap tide and are caused by surge (3.2 m during Idai; 2.6 m during Eloise) and wind setup (1.2 m during Idai; 0.7 m during Eloise). The maximum surge and its*

*timing during Idai (4.0 m) are in line with the operational forecast of 4.4 m based on the HyFlux2 model forced with NOAA Hurricane Weather Research and Forecast atmospheric data (ERCC, 2019; Probst and Annunziato, 2019). As tide and wave effects are not simulated by this model, total water levels are not available for comparison. In comparison with the tidal constituents of International Hydrographic Organization (IHO) station at the Port of Beira as retrieved using the Delft Dashboard (van Ormondt et al., 2020), the highest astronomical tide is expected to be around 3.8 m while our simulations result in 4.5 m, indicating an overestimation. This has however little effect on the maximum water levels which occurred close to neap tide.*

**3.2 Figure 3:** What does "actual event discharge" mean? The river discharge based on the gauge records or the CaMa discharge? If the latter, I would call it model-based discharge since it is not the "true" discharge.

Thanks for noticing this ambiguous wording. We used "actual event discharge" to distinguish it from the climatological discharge but agree that the wording is not clear. We have rephrased the caption to be more specific:

*L136: Figure 3: SFINCS boundary conditions during cyclone Idai (left column) and cyclone Eloise (right column) for total sea level from GTSM and ERA5 (top row); discharge from CaMa-Flood (center row); and spatial average runoff from ERA5 (bottom row). The full lines show the total water levels and discharge, as used for the validation, see Section 3.3.1. The dashed lines show the tidal water level component only (top row) and normalized discharge to match the climatological mean (center row), as used in the compound driver analysis, see Section 3.3.3. Only one coastal location (H4) and the two main rivers (Q1 - Buzi and Q4 - Pungwe) are shown to improve the readability of the plots. The location labels in the legends correspond to the locations as shown in Figure 2B.*

**3.3:** In addition to simulated flood extent, can any comparison be made using simulated vs satellite-based flood depth? In low-lying regions, the flood extent could be similar between the model and satellite, but the depth could be significantly different. I'm not asserting that the author's flood model is inaccurate, but just want to point out that a comparison based on flood extent alone does not provide a complete picture about whether flood dynamics are being accurately captured by the modeling framework.

Unfortunately, hardly any ground observations of water level / depth are available for our case study. We also checked if ICESAT-2 data are available for any of the events but found there is no overpass between 18-21 March 2019 and 24-27 Jan 2021. However, we do agree with the reviewer about the

need for flood depth observations and the potential for satellite-derived observations and discuss this in Section 4.1, 4.3 and in the conclusions in Section 5:

*L439: Our results also demonstrate that a commonly used metric to evaluate flood models such as the critical success index can mask large differences between model results and should be evaluated together with the false alarm and hit ratios and inspection of the geographical patterns and differences. An additional comparison with flood levels, if available, would allow for an even more comprehensive validation (Stephens and Bates, 2015; Wing et al., 2021).*

*L512: We recommend investigating whether remote sensing, e.g. satellite laser or radar altimetry data, can be used to validate extreme inland and nearshore water levels (Andreadis et al., 2020; O'Loughlin et al., 2016; Urban et al., 2008).*

*L545: We also reiterate the importance of observed water levels for a more comprehensive comparison of flood simulations.*

**4.1** It seems that although SFINCS simulates a larger extent of flooding than CaMa (due to incorporation of pluvial runoff), CaMa consistently predicts higher flood depths for both storms (except at location 5). I wonder if the authors have any ideas why CaMa would estimate higher flood depth than SFINCS?

In section 4.1 we argue that this is because of limited connectivity between neighboring floodplain cells in CaMa-Flood. CaMa-Flood is in essence a quasi 2D model as the main connection between cells is along the river network defined by a single downstream neighboring cell, while some water is exchanged with other adjacent cells through its bifurcation scheme (Yamazaki et al., 2014). This scheme is however too limited to represent the connectivity in the large low-gradient floodplains of the Buzi and Pungwe rivers. Therefore, more water is retained in the Pungwe in Buzi river cells compared to the SFINCS model. We have made some changes in section 4.1 to clarify this.

*L378: The difference between both models in the Buzi floodplains is likely due to the limited connectivity between floodplains of neighboring cells in the CaMa-Flood model through its so-called bifurcation schema. This scheme is however too limited to represent the connectivity in the large low-gradient floodplains of the Buzi and Pungwe rivers. This can be seen in the downscaled CaMa-Flood flood maps, which show unrealistic sudden local drops in flood depth at the interface of unit catchments during cyclone Idai (Figure 4A) and larger simulated water levels in the Buzi in CaMa-Flood compared to SFINCS (Figure 5A/B).*

**References**

Dullaart, J. C. M., Muis, S., Bloemendaal, N., and Aerts, J. C. J. H.: Advancing global storm surge modelling using the new ERA5 climate reanalysis, Clim. Dyn., 54, 1007–1021, https://doi.org/10.1007/s00382-019-05044-0, 2020.

Yamazaki, D., Sato, T., Kanae, S., Hirabayashi, Y., and Bates, P. D.: Regional flood dynamics in a bifurcating mega delta simulated in a global river model, Geophys. Res. Lett., 41, 3127–3135, https://doi.org/10.1002/2014GL059744, 2014.

---

## Author Response (AR2)

Dear editor,

Apologies for the miscommunication in the previous revision round and thanks for your useful comments. Please find a full response to both comments below. We have furthermore double checked the manuscript for typos and believe it's now ready for a final assessment by the editorial team.

Kind regards,
On behalf of all authors,
Dirk Eilander

*The two selected case studies are clearly relevant in terms of dynamics and intensity of the floods. However, they lack in situ observations of the water level. Please add in section 2 a short sentence to justify your choice in spite of this limitation. Could you have considered other locations where the flood analysis would have been supported by an in-situ observations?*

We have added a small paragraph to the end of section 2 to justify the selection of the case studies:

"These events were selected because they provide a unique case study of two different compound flood events in the same study area, allowing for a comparison of the compound flood dynamics between both events. Furthermore, the lack of compound flooding in global models has been identified as a key limitation to support decision making in this area (Emerton et al., 2020)."

*Referring to figure 2, you have maintained the terminology " model observation locations " that I suggested to replace with "model points". Please, explain why.*

We agree with the editor and have renamed "model observation points" to "model output points" in the legends of Figure 2 and 5 and any references to these figures.